# Trans-ancestry analysis in over 799,000 individuals yields new insights into the genetic etiology of colorectal cancer

**Changlong Yang**[1‡], **Zhenglin Chang**[2,3‡], **Youguo Dai**[1‡], **Jinzhao Mo**[5‡], **Qitai Zhang**[2‡], **Mingming Zhu**[1], **Likun Luan**[1], **Jinhu Zhang**[4*], **Baoqing Sun**[2,3*], **Junyi Jia**[1*]

**1** Department of Gastric and Intestinal Surgery, Yunnan Tumor Hospital, The Third Affiliated Hospital of Kunming Medical University, Kunming, China, **2** Department of Clinical Laboratory of the First Affiliated Hospital of Guangzhou Medical University, State Key Laboratory of Respiratory Disease, National Center for Respiratory Medicine, National Clinical Research Center for Respiratory Disease, Guangzhou Institute of Respiratory Health, Guangzhou, China, **3** Guangzhou Laboratory, Guangzhou International Bio Island, Guangzhou, Guangdong Province, China, **4** Department of Urology, Suizhou Central Hospital, The Fifth Affiliated Hospital of Hubei University of Medicine, Suizhou, Hubei, China, **5** Southern Medical University, Guangzhou, Guangdong Province, China

‡ CY, ZC, YD, JM and QZ contributed equally to this work and share first authorship.
* cmilyforyou@qq.com (JZ); sunbaoqing@vip.163.com (BS); 364929601@qq.com (JJ)

**Data Availability Statement:** The datasets analyzed during the current study are available in the IEU open gwas project, https://gwas.mrcieu.ac.uk. GWAS ID has been provided in S1-S2 Table,

## Abstract

### Background

Recent studies have demonstrated the relevance of circulating factors in the occurrence and development of colorectal cancer (CRC); however, the causal relationship remains unclear.

### Methods

Summary-level data for CRC were obtained from the UK Biobank (5,657 cases and 372,016 controls), FinnGen cohort (3,022 cases and 215,770 controls), and BioBank Japan Project (BBJ, 7,062 cases and 195,745 controls). Thirty-two peripheral markers with consistent definitions were collected from the three biobanks. Mendelian randomization (MR) was used to evaluate the causal effect of circulating factors on CRC. The effects from the three consortiums were combined using trans-ancestry meta-analysis methods.

### Results

Our analysis provided compelling evidence for the causal association of higher genetically predicted eosinophil cell count (EOS, odds ratio [OR], 0.8639; 95% confidence interval [CI] 0.7922–0.9421) and red cell distribution width (RDW, OR, 0.9981; 95% CI, 0.9972–0.9989) levels with a decreased risk of CRC. Additionally, we found suggestive evidence indicating that higher levels of total cholesterol (TC, OR, 1.0022; 95% CI, 1.0002–1.0042) may increase the risk of CRC. Conversely, higher levels of platelet count (PLT, OR, 0.9984; 95% CI, 0.9972–0.9996), total protein (TP, OR, 0.9445; 95% CI, 0.9037–0.9872), and C-reactive protein (CRP, OR, 0.9991; 95% CI, 0.9983–0.9999) may confer a protective effect against

and the direct link to each data can be obtained by searching for GWAS ID in the IEU website.

**Funding:** The authors received no specific funding for this work.

**Competing interests:** The authors have declared that no competing interests exist.

CRC. Moreover, we identified six ancestry-specific causal factors, indicating the necessity of considering patients' ancestry backgrounds before formulating prevention strategies.

## Conclusions

MR findings support the independent causal roles of circulating factors in CRC, which might provide a deeper insight into early detection of CRC and supply potential preventative strategies.

## Introduction

Colorectal cancer (CRC) is a major public health concern, ranking as the third most frequent cancer globally and second in terms of mortality [1]. Globally, the incidence of colorectal cancer cases more than doubled from 1990 to 2019, rising from 842,098 to 2.17 million, while deaths from this disease increased from 518,126 to 1.09 million [2]. In light of the heavy economic and medical burden of CRC, it is imperative to identify modifiable risk factors for primary prevention and to reduce the risk of cancer. Although genetics is a dominant risk factor, several lifestyle-related and demographic factors, such as inflammation, age, smoking, alcohol consumption, obesity, ancestry, and sex differences, are also known to increase the risk of CRC [3]. A better understanding of the causality and magnitude of these risk factors can significantly improve prevention strategies and offer new treatment targets.

Recently, a growing body of observational studies has revealed a correlation between various blood-based, heritable factors and disease, including eosinophil [4], neutrophil [5], basophil [6], platelet [7, 8], hematocrit [9], glucose [10], c-reactive protein [11], and dyslipidemia [12]. Despite evidence supporting the role of these factors in disease etiology, their causal relationship remains to be firmly established. Observational studies, particularly retrospective ones, are often subject to residual confounding bias and reverse causation, whereas randomized studies may also be limited in practice [13]. The Mendelian randomization (MR) approach offers a promising alternative for investigating the causal relationships between exposure and disease. MR is a widely accepted method for inferring causal relationships without the limitations of randomized clinical trials and is increasingly used in the field of medical research [13, 14]. Although recent MR studies have begun to investigate the links between modifiable factors and CRC [15, 16], the relationship between blood-based, heritable factors and disease remains an area ripe for further exploration.

Our study used a MR approach to investigate the causal relationship between 32 peripheral markers and CRC in a large sample of over 799,272 individuals from the UK Biobank, Finn-Gen, and BioBank Japan cohorts. The results of our study provide valuable information for understanding the causal relationships between peripheral markers and CRC and may have implications for the primary prevention of CRC. Further validation is required to confirm the reliability of our findings.

## Methods

### Study design

The MR design, an instrumental variable (IV) analysis, was applied to strengthen the inference of the causal effect of exposure on outcomes by exploiting genetic variants as IVs of exposure [17, 18]. Owing to such a re-analysis of publicly available summary-level data from large

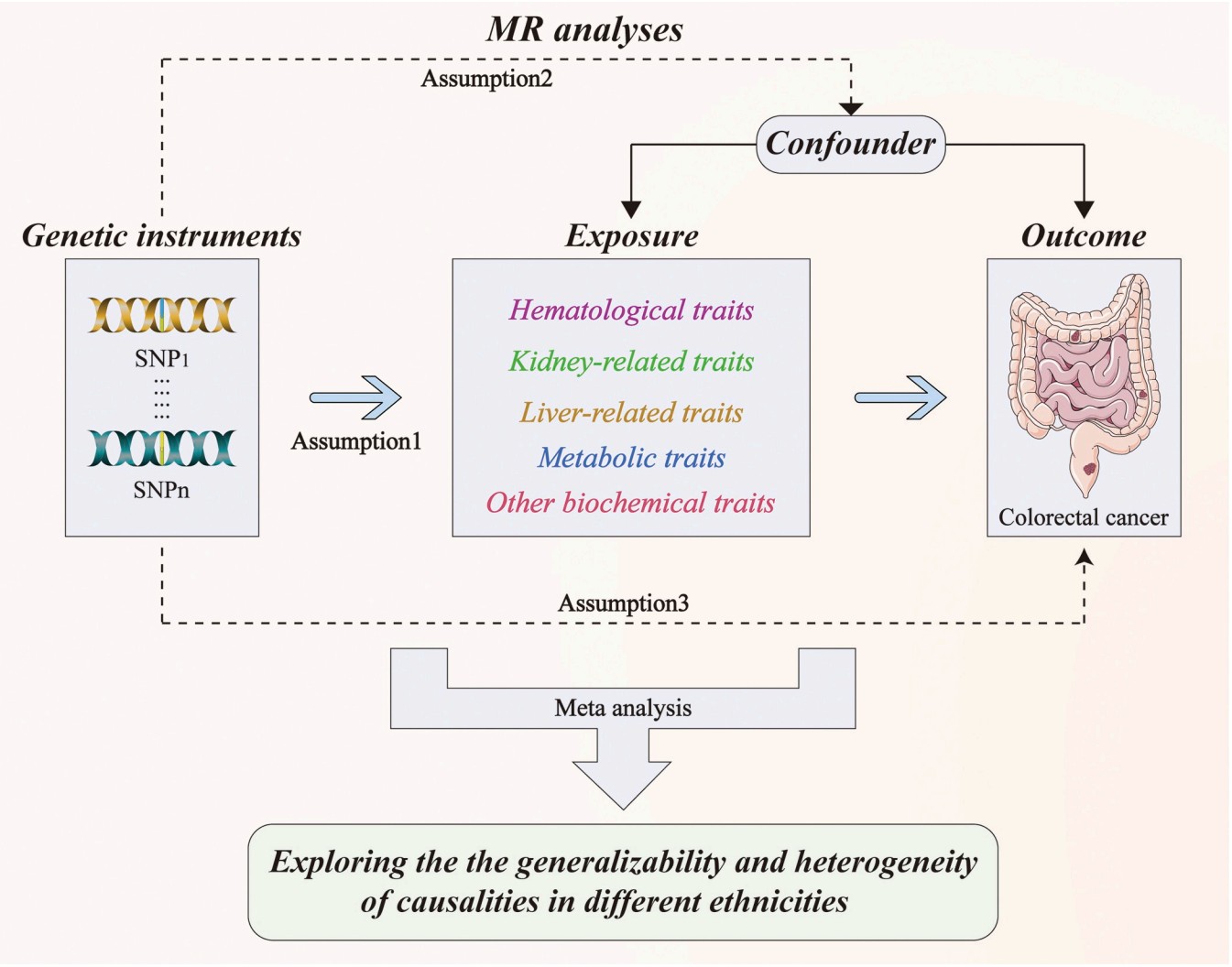

**Fig 1. Workflow of designed analysis.**

genome-wide association studies (GWAS), no additional ethical approval was required. Specifically, we described the MR analysis that we conducted to investigate the potential causal relationship between peripheral markers and the risk of CRC. Next, we conducted trans-ancestry meta-analyses of MR results for 32 peripheral markers in three biobanks to investigate whether these causal associations are shared across broad ancestries. Finally, we performed a meta-analysis of the causality of the remaining peripheral biomarkers in CRC to explore ancestry heterogeneity. The workflow is illustrated in **Fig 1**.

## Quality control

MR design was conducted according to three basic assumptions: the genetic variants selected as IVs should be robustly associated with the exposure (assumption 1); the selected variants should not be associated with any confounders (assumption 2); and the variants used should not be associated with the outcome, except by way of exposure (assumption 3). To ensure the integrity and reliability of our data, we implemented rigorous and comprehensive quality control (QC) procedures for each ancestry group. These procedures included restricting SNP

missingness to <0.05 and ensuring a minor allele frequency (MAF) of >0.01 [19]. Additionally, akin to methodologies employed in previous studies, quality assessment can be gauged by examining the adherence to the three stated assumptions [19–21].

## Collection of GWAS summary datasets

Summary-level data on CRC were derived from three nationwide biobanks (UK Biobank, FinnGen, and BioBank Japan). The sample size and proportion of cases are displayed in **Fig 2**. In UKB cohort, 5,657 cases (372,016 controls) of CRC were collected. In the FinnGen cohort, 3,022 cases (215,770 controls) with CRC were collected. In BBJ cohort, 7,062 cases (195,745 controls) with CRC were included. Summary-level statistical data for thirty-two peripheral markers with consistent definitions were collected from various sources. Similar to Saori et al.'s study [22], 32 peripheral biomarkers with consistent definitions from three biobanks were collected to explore the generalizability and heterogeneity of causalities of peripheral biomarkers among ancestries. When two or more exposures were presented in one article, each exposureoutcome association was taken as an independent study [16]. These factors were divided into five major groups: hematological, kidney-related, liver-related, metabolic, and other biochemical traits. Summary-level statistical European ancestry data for white blood cell count (WBC), basophil cell count (BASO), monocyte cell count (MONO), lymphocyte cell count (LYM), EOS, RDW, and hematocrit were downloaded from the Blood Cell Consortium [23, 24]. Summary-level statistical data for neutrophil count (NEU), red blood cell count (RBC), hemoglobin concentration (HGB), mean corpuscular hemoglobin concentration (MCHC), mean corpuscular hemoglobin (MCH), mean corpuscular volume (MCV), PLT, PDW, eGFRcys, Cr, alkaline phosphatase (ALP), alanine aminotransferase (ALT), aspartate aminotransferase (AST), gamma-glutamyltransferase (GGT), total bilirubin (Tbil), TP, glucose, HDL cholesterol (HDLC), LDL cholesterol (LDHC), TC, total protein (TG), CRP, hemoglobin A1c (HbA1c), and albumin (N ≤ 441,016) were obtained from the UK Biobank GWAS [25, 26]. Summary-level statistical mixed-population data for GFR were obtained from a large GWAS [27].

The summary-level statistical data for eGFR, sCr, ALP, ALT, AST, GGT, albumin (ALB), Tbil, TP, blood sugar, HDLC, LDHC, TC, TG, HbA1c, and CRP were obtained from a large GWAS that included 162,255 Japanese individuals [28]. East Asian ancestry summary-level statistical data for RDW and PDW were obtained from a multi-ancestry analysis across six continental ancestry groups. The summary-level statistical East Asian ancestry data for WBC, BASO, MONO, LYM, EOS, NEU, RBC, HGB, hematocrit (HCT), MCHC, MCH, and MCV were obtained from a large GWAS in 746,667 individuals from five global populations [23].

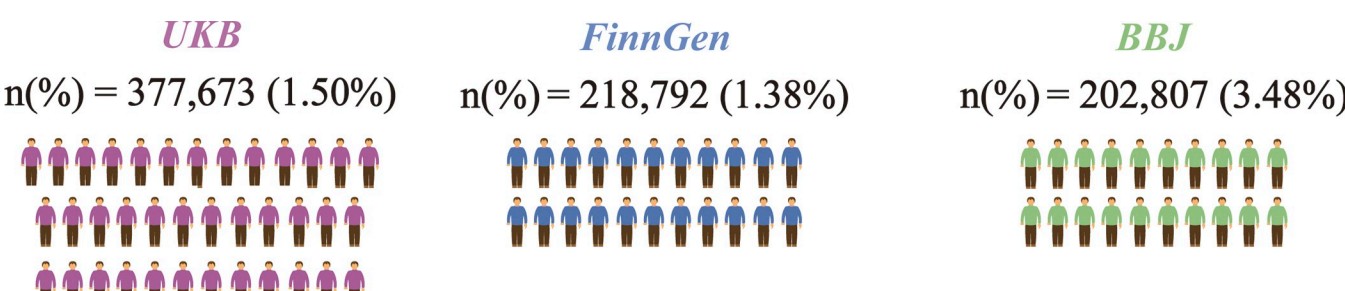

**Fig 2. The geographical locations of three cohorts and summarization of the sample size (the proportion of cases) of MR studies.**

## Genetic instrument of potential risk factors

A series of quality control steps were conducted to select eligible instrumental single-nucleotide polymorphisms (SNPs). First, SNPs robustly associated with exposure at the genome-wide significance level ($P < 5 \times 10^{-8}$) were selected. Second, we applied a linkage disequilibrium (LD) clumping process (LD $r^2 < 0.001$, clumping distance cut-off = 10,000kb) to estimate the LD between SNPs. To minimize pleiotropy, Phenoscanner2 (http://www.phenoscanner.medschl.cam.ac.uk) was used to determine whether any exposure-associated SNPs were associated with relevant confounders in CRC, and their specific details have been included in S14 Table in S4 File. Subsequently, the SNPs that were unavailable in the outcome GWAS were removed, and the extracted SNPs were not significantly associated with the outcome ($P > 5 \times 10^{-5}$). Harmonization was applied to exclude palindromic and incompatible SNPs, particularly focusing on removing palindromic SNPs with intermediate allele frequencies (40–70%). For SNPs exhibiting pleiotropy, we adopted a conservative approach and excluded them from our analysis. Finally, to test whether there was a weak IV bias, namely, genetic variants selected as IVs having a weak association with exposure, we calculated the F statistic ($F = R^2/(1 − R^2) \times (n − k − 1)/k$; n, sample size; k, number of instrumental variables; and MAF, minor allele frequency) [29]. The variance ($R^2$) represents the phenotype variance induced by the SNPs [30]. When $R^2$ is not available, we use the formula $R^2 = 2 \times MAF \times (1-MAF) \times Beta^2$ (where beta represents the effect value of the genetic variant in the exposure.

## Statistical analyses

The "TwoSampleMR" package based on R 4.03 was applied to perform MR analysis. After harmonization of the effect alleles across the GWAS of exposures and urolithiasis, the conventional fixed-effects inverse-variance weighted (IVW) method was used as the main statistical model of causal estimates. The estimates were combined using a meta-analysis. A fixed-effects model was used when heterogeneity was low ($I^2$ values were lower than 50%); otherwise, a random-effects model was used. The IVW method assumes that instruments can affect the outcome only through the exposure of interest and not through any alternative pathway [31]. Moreover, we employed MR-Egger and weighted median analyses alongside IVW to assess the presence of directional pleiotropy, which may suggest collider bias [32, 33]. The MR-Egger test for directional pleiotropy and Cochran's Q statistics were used to identify the presence of significant heterogeneity or directional pleiotropy. To address multiple testing, a conservative Bonferroni-corrected threshold ($P < 0.05/32 = 1.56 \times 10^{-3}$, because 32 factors were evaluated for MR analysis. P values that remain significant after Bonferroni correction provide compelling evidence, while P values in the range of 0.00156 to 0.05 suggest suggestive evidence.

## Results

### Information on outcomes and exposures

Three biobanks were included in our analyses (Table 1). In BBJ cohort, 7,062 cases (195,745 controls) with CRC were included. Subsequently, we collected 32 peripheral risk factors with a consistent definition from the two cohorts (Fig 3). The genetic instrument of peripheral markers in European ancestry ranged from 46 to 510 (S1 Table in S1 File), and IVs of peripheral markers in East Asian ancestry ranged from 5 to 142 (S2 Table in S1 File). Nearly, all these exposures had strong genetic instruments (F statistics > 10 for 62 of the 64 selected markers; S1 and S2 Tables in S1 File). Detailed information on CRC-independent SNPs (after the clumping process) for peripheral risk factors is listed in S3-S7 Tables in S2 File.

**Table 1. Description of colorectal cancer outcomes used in this study.**

| Trait_name | N_case | N_control | Sample_size | Consortium |
|---|---|---|---|---|
| Colorectal cancer | 5,657 | 372,016 | 377,673 | UKB |
| Colorectal cancer | 3,022 | 215,770 | 218,792 | Finngen |
| Colorectal cancer | 7,062 | 195,745 | 202,807 | BBJ |

### Trans-ancestry meta-analysis of CRC in three nationwide biobanks

To investigate whether these causal associations are shared across broad ancestries, we conducted trans-ancestry meta-analyses of MR results of 32 peripheral markers in three biobanks. Among the 32 biomarkers, higher levels of TC were significantly associated with a higher risk of CRC, while higher levels of EOS, RDW, PLT, TP, and CRP were significantly associated with a lower risk of CRC (**Fig 4A and 4B**). ORs of CRC were 1.0022 (95% CI, 1.0002–1.0042, p = 0.030) per a 1-SD increase in TC levels, 0.8639 (95% CI, 0.7922–0.9421, P < 0.001) per a

## 32 peripheral markers with consistent definition from three cohorts

**Trait_name**

### Hematological traits

*White blood cell count (WBC)*

*Eosinophil cell count (EOS)*

*Basophil count (BASO)*

*Lymphocyte cell count (LYM)*

*Monocyte cell count (MONO)*

*Neutrophil cell count (NEU)*

*Red blood cell count (RBC)*

*Hemoglobin concentration (HGB)*

*Hematocrit (HCT)*

*Red cell distribution width (RDW)*

*Mean corpuscular hemoglobin (MCH)*

*Mean corpuscular volume (MCV)*

*Mean corpuscular hemoglobin concentration (MCHC)*

*Platelet count (PLT)*

*Platelet distribution width (PDW)*

### Kidney-related traits

*Glomerular filtration rate (GFR)*

*Serum cystatin C (eGFRcys)*

*Serum creatinine (sCr)*

**Trait_name**

### Liver-related traits

*Alkaline phosphatase (ALP)*

*Alanine aminotransferase (ALT)*

*Aspartate aminotransferase (AST)*

*Gamma glutamyltransferase (GGT)*

*Albumin (ALB)*

*Total bilirubin (Tbil)*

*Total protein (TP)*

### Metabolic traits

*Blood sugar/Glucose*

*High-density-lipoprotein cholesterol /HDL cholesterol (HDLC)*

*Low-density-lipoprotein cholesterol /LDL cholesterol (LDLC)*

*Total cholesterol (TC)*

*Triglyceride (TG)*

*Hemoglobin A1c (HbA1c)*

### Other biochemical traits

*C-Reactive protein (CRP)*

**Fig 3. The peripheral markers were selected from three biobanks.**

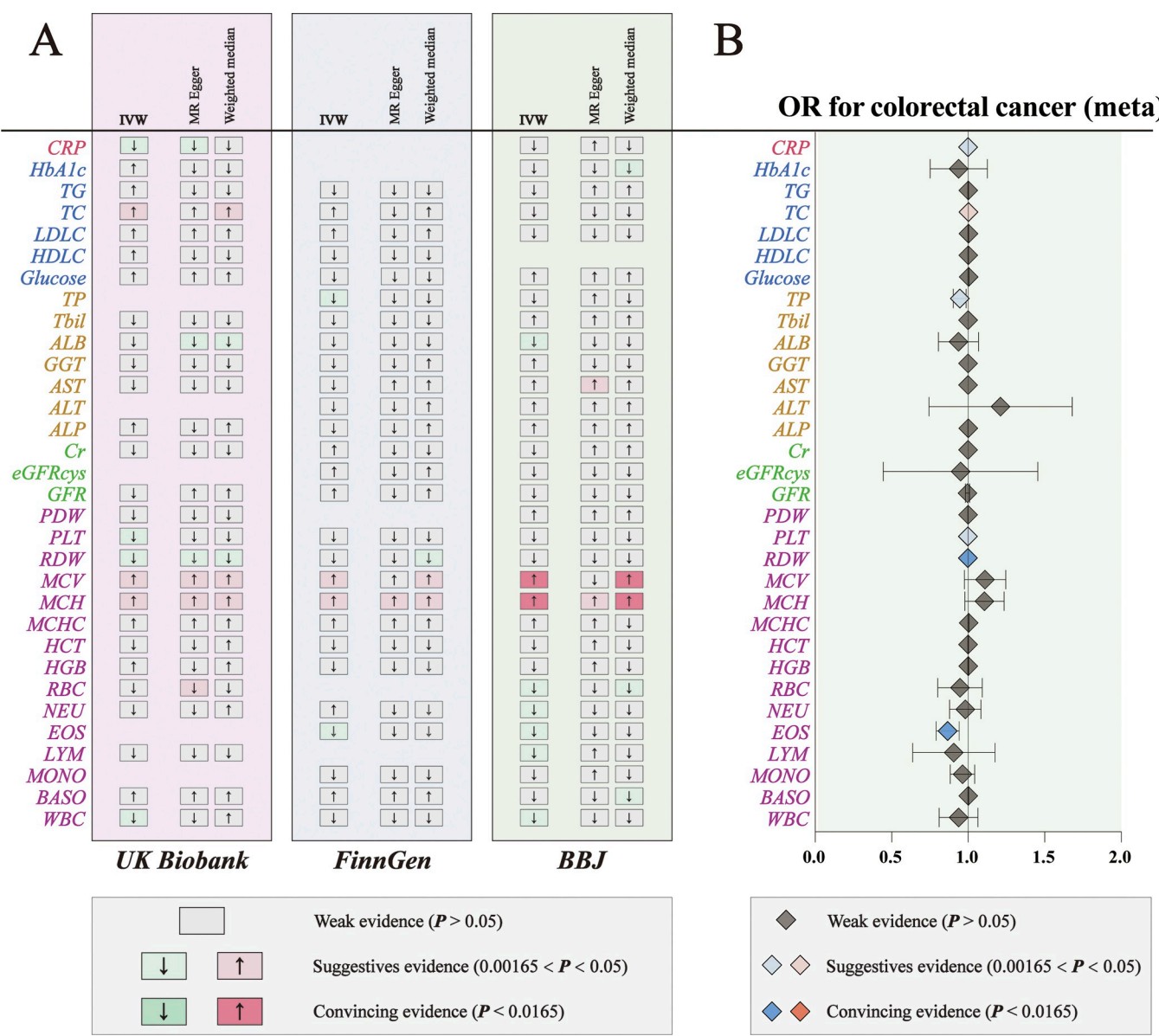

**Fig 4. Trans-ancestry association study of 32 common peripheral factors with colorectal cancer (CRC).** (A) Overview of the associations of 32 common peripheral markers on CRC. (B) The trans-ancestry meta-analysis of the association results from the three cohorts by the inverse-variance weighted method. The diamonds refer to the point estimates, and the horizontal bars represent the 95% confidence interval. The effect on the x-axis is the odds ratio of CRC per 1 SD change in the exposure.

1-SD increase in EOS levels, 0.9981 (95% CI, 0.9972–0.9989, P < 0.001) per a 1-SD increase in RDW levels, 0.9984 (95% CI, 0.9972–0.9996, p = 0.008) per 1-SD increase in PLT levels, 0.9445 (95% CI, 0.9037–0.9872, p = 0.010) per a 1-SD increase in TP levels, and 0.9991 (95% CI, 0.9983–0.9999, p = 0.030) per 1-SD increase in CRP levels. These findings provide compelling evidence for the impact of EOS and RDW levels on CRC risk, and suggestive evidence for the association between the levels of TC, PLT, TP, and CRP and CRC risk. In addition to this, MR-Egger and weighted median analyses also evaluated the heterogeneity and pleiotropy in our MR analysis (**S8-S10 Tables in S3 File and S11 Table in** S4 **File**). For MR results

exhibiting pleiotropy, we conducted exclusions (**Fig 4A**). These six heritable factors may reflect a shared genetic basis for CRC in both European and East Asian populations.

## Exploring the causal effects of ancestry heterogeneity on CRC

We then performed an ancestry heterogeneity meta-analysis of the causality of the remaining 26 peripheral biomarkers in CRC Among the remaining 26 biomarkers, higher WBC levels were significantly associated with a lower risk of CRC in European ancestry (**Fig 5A**). ORs of CRC were 0.9986 (95% CI, 0.9971–1.0000, P = 0.040) per 1-SD increase in WBC levels in European ancestry (**S12 Table in** S4 **File**). In East Asian ancestry, we observed that higher levels of MCH and MCV were significantly associated with a higher risk of CRC, while higher levels of WBC, LYM, RBC, and ALB were significantly associated with a lower risk of CRC (**Fig 5B**). The ORs of CRC were 1.2102 (95% CI, 0.57–0.79, P < 0.001) per a 1-SD increase in MCH levels, 1.2314 (95% CI, 1.1180–1.3563, P < 0.001) per a 1-SD increase in MCV levels, 0.7863(95% CI, 0.6670–0.9269, P = 0.004) per a 1-SD increase in WBC levels, 0.7313 (95% CI, 0.5629–0.9500, P = 0.019) per a 1-SD increase in LYM levels, 0.8494(95% CI, 0.7270–0.9924, P = 0.040) per a 1-SD increase in RBC levels, and 0.7384 (95% CI, 0.5874–0.9283, P = 0.009) per a 1-SD increase in ALB levels. These findings provide compelling evidence for the association of MCH and MCV levels with CRC risk, and suggestive evidence for the association of WBC, LYM, RBC, and ALB levels with CRC risk in East Asian ancestry. In European ancestry, we found suggestive evidence for the association between WBC levels and CRC risk. Detailed results are provided in **S12 and S13 Tables in** S4 **File**.

## Discussion

In this study, we aimed to develop strategies for CRC prevention by identifying and comparing causal risk factors across different populations. To achieve this goal, we conducted a Mendelian randomization analysis using summary-level data from three large biobanks (UK Biobank, FinnGen, and BioBank Japan). Our analysis was based on 799,272 participant samples and 32 peripheral markers, with selected SNPs serving as exposures.

Our study found that six heritable causal factors were consistently detected in both ancestries. Tumor immunosurveillance suggests that CRC, which is characterized by high levels of circulating EOS, can reduce the risk of cancer by enhancing the immune system's ability to detect and eliminate malignant cells [4, 34]. Our study found that EOS was the strongest causal marker among the 32 peripheral markers associated with CRC in our trans-ancestry meta-analysis. In agreement with the results of MR analysis, our observational data also support the important role of EOS as a clinical marker of CRC. Multiple parallel pathways have been shown to be involved in eosinophil infiltration in colorectal tumors and the direct killing of malignant cells [35–37]. These phenomena have been experimentally demonstrated. Enhanced Th2 immune response, certain components of Th2-driven inflammation in cancer, may be associated with the antitumor activity of CD4 Th2 cells and tumor-infiltrating granulocytes, especially regulatory eosinophils [35, 36]. The expression of IL33 by tumor cells induces the production of eotaxin 1, leading to eosinophil recruitment and degranulation-dependent suppression of tumor growth [37]. Clearly, the antitumor effect of eosinophils in vivo may involve multiple interactions with other immune cells [35, 36].

The relationship between CRP and the risk of CRC has been a subject of debate in the scientific community. Some studies suggest that CRP, as a biomarker of low-grade inflammation, may play a role in CRC, whereas others have found mixed or null results [38, 39]. Chronic inflammation may lead to reactive oxygen and nitrogen species released by inflammatory cells, resulting in malignant DNA mutations [11]. As a biomarker of low-grade inflammation, CRP

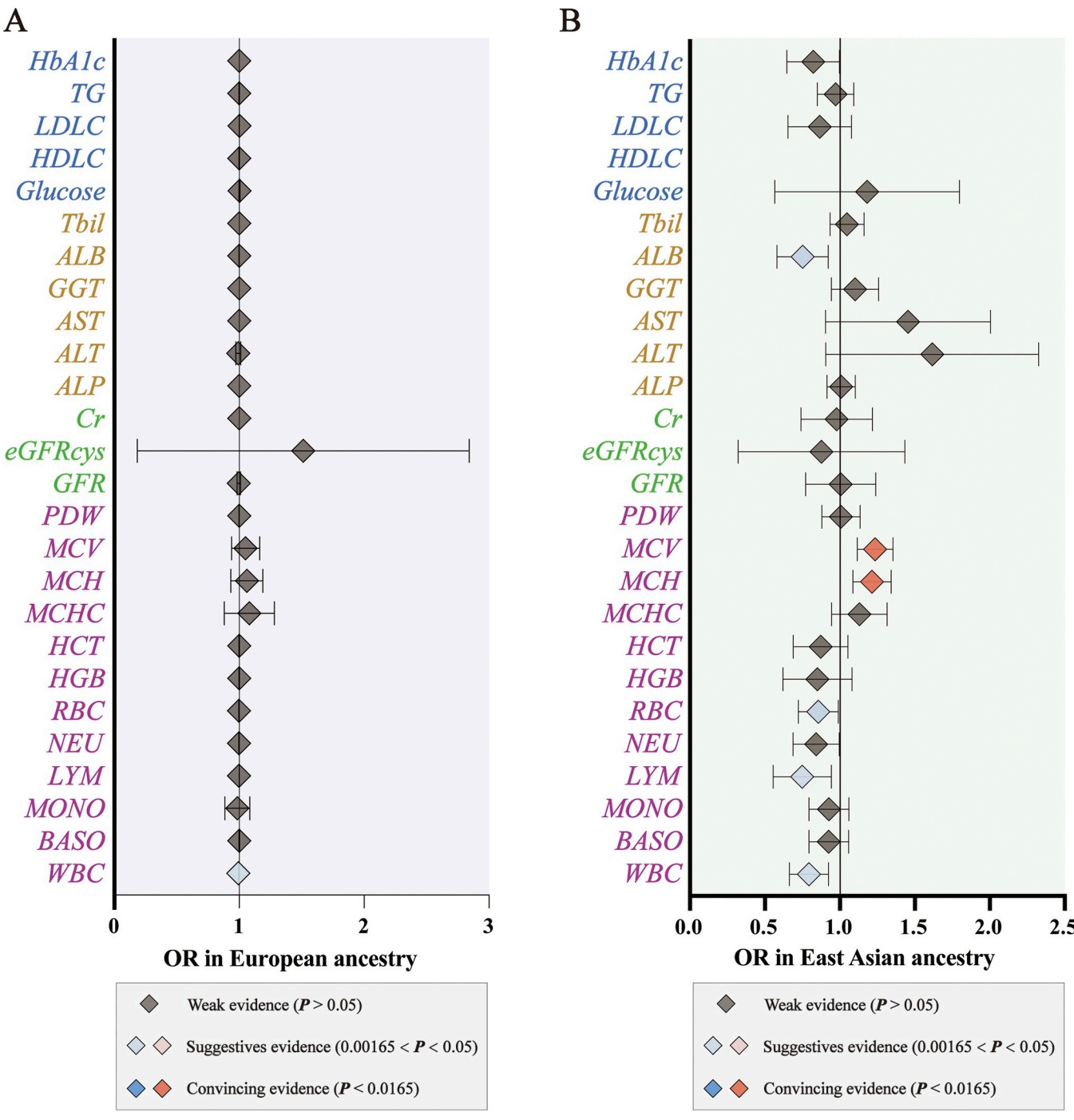

**Fig 5. Ancestry heterogeneity meta-analysis of the remaining 26 peripheral biomarkers in CRC. (A-B)** Forest plot for causal effects of the rest of 26 risk factors on CRC in European and East Asian ancestry. The diamonds refer to the point estimates, and the horizontal bars represent the 95% confidence interval. The effect on the x-axis is the odds ratio of CRC per 1 SD change in the exposure.

has been proposed to play a role in CRC [40]. According to three studies, there were positive associations with relative risks (RRs) of 1.4, 2.0, and 2.9 [41–43]. One group found an inverse relationship between proximal colon cancer and proximal colon cancer, with four reporting null findings [44–47]. Our study demonstrated an inverse relationship between CRP level and

CRC. This is inconsistent with previous literature findings. To address this phenomenon, we consider the possibility of confounding factors. Many previous studies have reported systemic inflammatory markers that can be used to predict prognosis, such as the neutrophil-lymphocyte ratio (NLR), the lymphocyte-monocyte ratio (LMR), the platelet-lymphocyte ratio (PLR), the Glasgow prognostic score (GPS), lymphocyte-C-reactive protein ratio (LCR), systemic inflammation score (SIS), and prognostic nutritional index (PNI). Compared with traditional individual inflammatory markers, combining these parameters is more effective in predicting cancer prognosis. However, based on the Mendelian randomization principle, these markers are not applicable [48, 49]. LMR is an independent predictor of OS in patients with CRC undergoing curative resection and appears superior to preexisting biomarkers [50]. However, further research is needed to fully understand the mechanisms underlying these associations. Further genetic and clinical studies are needed to clarify the relationship between CRP and the risk of CRC.

Our research demonstrated a clear positive correlation between elevated serum TC levels and an increased risk of CRC. The mechanisms underlying this relationship are still being explored. Our meta-analysis provides evidence of a positive association between TC levels and the prevalence of CRC, with one prospective study indicating a higher risk of colon cancer than rectal cancer [51]. Potential factors that contribute to this relationship include genetic influences such as the apoE phenotype, which affects both cholesterol metabolism and susceptibility to carcinoma [52]. Observational studies have also suggested that statins may reduce the risk of advanced adenomatous polyps and CRC [53–55]. Studies have revealed that immune cells, such as activated T cells, undergo rapid proliferation during tumor immunity and require a significant amount of cholesterol. Cholesterol is strongly attracted to Dishevelled (DVL), a protein that acts as a scaffold in the Wnt pathway. The presence of cholesterol activates canonical Wnt signaling by enhancing the membrane recruitment of DVL and facilitating its interaction with Frizzled (FZD) and low-density lipoprotein receptor-related protein 5/6 (LRP5/6) [56]. Intracellular cholesterol can activate the MAPK pathway by increasing reactive oxygen species (ROS) [57]. An animal study demonstrated that anaerobic digesting Streptococcus induces intracellular cholesterol biosynthesis in colonic cells, resulting in increased proliferation and impaired colonic cell development in mice [58]. Further comprehensive studies are required to fully elucidate this issue. Our study also supports a positive association between TC and CRC, highlighting the need for additional research in this area.

The role of hematological traits in the development of CRC has been extensively studied in recent years, focusing on the differential effects of peripheral markers, such as PLT, RBC, and WBC [59]. A case-control study revealed that univariate analysis showed a significant association between the CRP/MCV ratio, CRP level, and poor prognosis in CRC [60]. Tumor cell invasion of intestinal epithelial cells can lead to malnutrition-related chronic anemia, resulting in elevated MCV, MCH, and RDW. These hematological parameters undergo significant changes shortly before the diagnosis of CRC [61]. Our study found evidence for the protective effect of higher hematocrit levels on CRC and also revealed the direct causal nature of platelets and RDW on CRC in both ancestries. Causal relationships between the remaining hematological markers and CRC were only observed in East Asians. This highlights the importance of further exploring the causal mechanisms underlying these associations and their potential use as clinical markers for CRC risk assessment.

In conclusion, our study provides evidence for the causal role of several peripheral markers, including EOS, CRP, TC, TP, PLT, RDW, WBC, ALB, MCV, MCH, RBC, and LYM, in developing CRC. Although we conclude no causal associations for the remaining 20 markers, these negative results are as important as the positive discoveries for a fuller picture of the complicated etiology of CRC. These results may have important implications in the prevention and

early detection of CRC. Furthermore, our findings reveal significant differences in CRC markers between European and East Asian populations. These variances are likely attributable to an intricate array of ancestry-specific genetic traits, environmental factors, and lifestyle practices [62]. Recognizing these differences is essential for developing targeted strategies in CRC prevention and early detection, and emphasizes the value of personalized healthcare.

However, it is important to acknowledge the limitations of our study. Firstly, despite employing rigorous methods to filter potential confounders, the differentiation between confounding and mediating variables may still present some limitations. Additionally, although we made efforts to ensure distinct datasets from different sources and conducted extensive sensitivity analyses to assess the robustness of our MR estimates in the presence of any potential overlap, we recognize that there may still be some impact on our results due to the presence of overlap. A limitation in our sensitivity analysis was the absence of machine learning evaluations with and without ambiguous SNPs. Furthermore, not performing Principal Component Analysis (PCA) for ancestry could have introduced heterogeneity in the European descent group, potentially affecting genetic association signals. Lastly, the influence of reverse causation should not be disregarded [63, 64].

## Supporting information

**S1 File.**
(XLSX)

**S2 File.**
(XLSX)

**S3 File.**
(XLSX)

**S4 File.**
(XLSX)

## Acknowledgments

We thank Home for Researchers editorial team (www.home-for-researchers.com) for language editing service.

## Author Contributions

**Conceptualization:** Zhenglin Chang.

**Data curation:** Youguo Dai, Likun Luan, Jinhu Zhang.

**Formal analysis:** Changlong Yang, Qitai Zhang.

**Funding acquisition:** Changlong Yang, Junyi Jia.

**Methodology:** Zhenglin Chang.

**Project administration:** Youguo Dai.

**Resources:** Youguo Dai, Junyi Jia.

**Software:** Mingming Zhu.

**Supervision:** Zhenglin Chang, Baoqing Sun, Junyi Jia.

**Validation:** Mingming Zhu, Likun Luan.

**Visualization:** Jinzhao Mo.

**Writing – original draft:** Changlong Yang, Jinhu Zhang.

**Writing – review & editing:** Jinzhao Mo, Qitai Zhang, Likun Luan, Baoqing Sun, Junyi Jia.

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
