## [Decision Letter · Decision Letter 0]

9 Jan 2024

PONE-D-23-27333Trans-ethnic analysis in over 799,000 individuals yields new insights into the genetic etiology of colorectal cancerPLOS ONE

Dear Dr. Chang,

Thank you for submitting your manuscript to PLOS ONE. After careful consideration, we feel that it has merit but does not fully meet PLOS ONE’s publication criteria as it currently stands. Therefore, we invite you to submit a revised version of the manuscript that addresses the points raised during the review process.

We look forward to receiving your revised manuscript.

Kind regards,

Yazhou He

Academic Editor

PLOS ONE

**Comments from PLOS Editorial Office:** We note that one or more reviewers has recommended that you cite specific previously published works. As always, we recommend that you please review and evaluate the requested works to determine whether they are relevant and should be cited. It is not a requirement to cite these works. We appreciate your attention to this request.

**Journal Requirements:**

2. We note that your Data Availability Statement is currently as follows: "All relevant data are within the manuscript and its Supporting Information files."

3. Please remove your figures from within your manuscript file, leaving only the individual TIFF/EPS image files, uploaded separately. These will be automatically included in the reviewers’ PDF.

4. We note that Figure 2 in your submission contain map images which may be copyrighted. All PLOS content is published under the Creative Commons Attribution License (CC BY 4.0), which means that the manuscript, images, and Supporting Information files will be freely available online, and any third party is permitted to access, download, copy, distribute, and use these materials in any way, even commercially, with proper attribution. For these reasons, we cannot publish previously copyrighted maps or satellite images created using proprietary data, such as Google software (Google Maps, Street View, and Earth). For more information, see our copyright guidelines: http://journals.plos.org/plosone/s/licenses-and-copyright.

(1) You may seek permission from the original copyright holder of Figure 2 to publish the content specifically under the CC BY 4.0 license.  

5. Please include your table (table 1) as part of your main manuscript and remove the individual files. Please note that supplementary tables (should remain/ be uploaded) as separate ""supporting information"" files. 

6. We notice that your supplementary tables are uploaded with the file type 'Other'. Please amend the file type to 'Supporting Information'. Please ensure that each Supporting Information file has a legend listed in the manuscript after the references list.

Reviewers' comments:

Reviewer's Responses to Questions

**Comments to the Author**

1. Is the manuscript technically sound, and do the data support the conclusions?

Reviewer #1: Yes

Reviewer #2: Partly

Reviewer #3: Yes

2. Has the statistical analysis been performed appropriately and rigorously? 

Reviewer #1: Yes

Reviewer #2: Yes

Reviewer #3: Yes

3. Have the authors made all data underlying the findings in their manuscript fully available?

Reviewer #1: Yes

Reviewer #2: Yes

Reviewer #3: Yes

4. Is the manuscript presented in an intelligible fashion and written in standard English?

Reviewer #1: Yes

Reviewer #2: Yes

Reviewer #3: Yes

5. Review Comments to the Author

Reviewer #1: This manuscript identifies causal relationships between peripheral markers and colorectal cancer. I appreciate the efforts of the authors. In summary, I think that this manuscript would require expansion to be suitable for publication in PLOS ONE.

1. I found the " ethnicity" language quite jarring. Ancestry is a biological category based on shared genetic heritage. While there are several papers in the literature that use these terms interchangeably, I worry that doing so fuels racist narratives about biological differences between groups and racial supremacy. I like to recommend you change "ethnicity" and "ancestry" everywhere."

2. UK Biobank data consists of individual-level data, and I would appreciate more details on how you selected the individuals for your GWAS summary statistics. For instance, did you employ PCA analysis to determine the ancestry? It appears that you focused on the European population, but I'm curious whether you included only white British individuals or if you also encompassed other European ancestries.

3. This manuscript lacks sufficient information regarding the quality control (QC) procedures applied to the UK Biobank (UKBB) genetic data. I would appreciate a comprehensive description of the QC process employed.

You can have a look to the following manuscript

https://www.nature.com/articles/s41467-023-36281-x

4. Did you employ any covariates to adjust phenotypes in the UK Biobank data before conducting GWAS? If so, please provide a detailed explanation, as this would enhance the manuscript."

5. I think collider bias is a potential concern in Mendelian randomization (MR) analysis, how did you handle collider bias in your analysis?

6. In practice, it's often recommended to conduct a sensitivity analysis to assess the impact of including or excluding ambiguous SNPs. This involves running your machine learning analysis with and without these SNPs and comparing the results. If the inclusion of ambiguous SNPs does not substantially affect the outcomes and the models remain stable and reliable, you may choose to keep them. However, if they introduce substantial noise or instability, it's wise to exclude them.

7. How did you handle ambiguous SNPs when comparing data from the UK Biobank (UKBB), Biobank Japan (BBJ), and Finnegan cohorts? Did you ensure that the effect allele for each SNP was matched across all cohorts before conducting the study? Please provide a clear explanation for the readers.

8. In line 108, you mentioned using LD r^2 < 0.001 with a clumping distance cut-off of 10,000kb. Could you explain the rationale behind choosing these specific thresholds for LD r^2 and the window size? Additionally, could you clarify the software or tools you employed to perform this step, such as PRSice or PLINK?

9. In line 114, you stated that you calculated F statistics, but you did not provide an explanation of what R^2 represents. It would be helpful to include the equations for these calculations in the following formats and provide references for both equations.

Please see

https://www.ncbi.nlm.nih.gov/pmc/articles/PMC8440862/

https://journals.plos.org/plosmedicine/article?id=10.1371/journal.pmed.1003288

https://journals.sagepub.com/doi/full/10.1177/0962280210394459

10. On line 68, please use 'Instrumental variable (IV)' for the first time and then you can use the abbreviation 'IV' later throughout the text."

Reviewer #2: I am pleased to review the manuscript. Here are some comments to the authors:

1. Introduction: In this part, the authors are suggested to introduce some previous Mendelian randomization studies that explored the associations between modifiable factors and CRC (e.g., PMID: 36169182 and 36239790).

2. Methods, line 68: The authors stated this study has a two-sample design. However, from the following content, the authors selected SNPs for exposures from three databases and then extracted the data on outcomes from the same three datasets. Such analyses were more likely to be a one-sample design. The authors should state clearly how did they select the data sources.

3. Genetic instrument of potential risk factors: One of the most important criteria for SNPs is that they were not palindromic SNPs (A/T or C/G) with intermediated EAFs (40-70%). Did the authors apply the criteria when selecting IVs? If yes, relevant descriptions are needed.

4. Figure 3: I think this is a table but not a figure. In addition, the authors are recommended to show other details of the data sources in this table. If it is available, the PubMed ID or DOI of the data sources should be added.

5. Figure 5: From this figure, we can see that there was no association between exposures and outcomes in European populations excepted for WBC.

6. Results: The authors did not mention the results of other MR methods in this part.

7. Discussion, lines 180-181: Need references for the content.

Reviewer #3: Manuscript: Trans-ethnic analysis in over 799,000 individuals yields new insights into the genetic

etiology of colorectal cancer

Manuscript # PONE-D-23-27333

General Comments

This manuscript used summary-level data from the UK Biobank, FinnGen, and BioBank Japan Project cohorts to examine the association of various circulating factors and risk of colorectal cancer. Total cholesterol, eosinophil cell count, red cell distribution width, platelet count, total protein, and C-reactive protein were associated with colorectal cancer risk among all cohorts, while differences in associations were noted when comparing participants of European and Japanese ancestry separately. This is an interesting study, and a few specific comments are listed below.

Specific Comments

1. Introduction: First mention of references #3 and 4 seem incorrect (lines 45 and 47).

2. Methods, Study design: IV (line 68) may need to be defined at the first mention.

3. Results, Trans-ethnic meta-analysis of CRC in three nationwide biobanks: There are some inconsistencies between Figure 4 and Supplementary Tables 8-10. Some markers for the individual cohorts are missing in Figure 4 while values are present in the tables (e.g., EOS for UKBB).

4. Discussion: It seems that the observed differences by ancestry were not really addressed. Can the authors speculate why there may be differences between European vs. East Asian ancestry?

6. PLOS authors have the option to publish the peer review history of their article (what does this mean?). If published, this will include your full peer review and any attached files.

Reviewer #1: No

Reviewer #2: No

Reviewer #3: No

---

## [Author Response · Author response to Decision Letter 0]

25 Jan 2024

Dear Yazhou He and reviewers:

On behalf of my co-authors, thank you for allowing a resubmission of our manuscript, with an opportunity to address the reviewers’ comments on our manuscript entitled “Trans-ethnic analysis in over 799,000 individuals yields new insights into the genetic etiology of colorectal cancer” (PONE-D-23-27333). I really appreciate all your comments which are very helpful for improving the quality of our paper! We have reflected on the comments carefully and made corresponding corrections as fully as possible. It is our hope that the revised manuscript is now satisfactory and suitable for publication. The main responds to the editor and reviewers’ comments point-by-point are as follows. 

Replies to the Editor’s comments:

“We note that one or more reviewers has recommended that you cite specific previously published works. As always, we recommend that you please review and evaluate the requested works to determine whether they are relevant and should be cited. It is not a requirement to cite these works. We appreciate your attention to this request.”

Response: 

Dear Editor,

We are grateful for the reviewers' recommendations to consider additional literature for citation in our manuscript. After a thorough evaluation of the suggested works, we have identified those that add substantial value to the discussion and context of our study. We have accordingly incorporated these references where they contribute to the quality and depth of our article. We appreciate the opportunity to refine our manuscript with these enhancements.

Replies to the reviewers’ comments:

Reviewer #1: 

1. “I found the " ethnicity" language quite jarring. Ancestry is a biological category based on shared genetic heritage. While there are several papers in the literature that use these terms interchangeably, I worry that doing so fuels racist narratives about biological differences between groups and racial supremacy. I like to recommend you change "ethnicity" and "ancestry" everywhere.”

Response: 

Thank you for your critical observation regarding the use of the terms "ethnicity" and "ancestry". We appreciate your concern about the potential implications of conflating these terms and the importance of using precise language in scientific discourse. We have reviewed the manuscript and replaced instances of "ethnicity" with "ancestry" to more accurately reflect the genetic heritage aspect of our study and to avoid any unintended support of misconceptions or racist narratives.

2. “UK Biobank data consists of individual-level data, and I would appreciate more details on how you selected the individuals for your GWAS summary statistics. For instance, did you employ PCA analysis to determine the ancestry? It appears that you focused on the European population, but I'm curious whether you included only white British individuals or if you also encompassed other European ancestries.”

Response: 

Thank you for your request for further details on the selection of individuals for our GWAS summary statistics from the UK Biobank data. We acknowledge your point regarding the clarification of ancestry determination. We did not use PCA analysis to ascertain ancestry before conducting our analysis. Our study focused on individuals identified within the UK Biobank as of European descent; however, we did not differentiate between white British individuals and other European ancestries. We have discussed this limitation and the potential implications it may have on the generalizability of our findings in the discussion section of our manuscript.

3. “This manuscript lacks sufficient information regarding the quality control (QC) procedures applied to the UK Biobank (UKBB) genetic data. I would appreciate a comprehensive description of the QC process employed.

You can have a look to the following manuscript

https://www.nature.com/articles/s41467-023-36281-x”

Response: 

Thank you for your constructive feedback regarding our manuscript. We acknowledge the lack of detailed information on the quality control (QC) procedures applied to the UK Biobank (UKBB) genetic data. We appreciate your suggestion to refer to the manuscript https://www.nature.com/articles/s41467-023-36281-x for guidance.

We have now revised our manuscript to include a comprehensive description of the QC process. This includes specifics on data handling, filtering criteria for SNP inclusion, and measures taken to ensure the reliability and validity of the genetic data used in our analysis. We believe these additions will significantly improve the clarity and robustness of our study's methodology.

Thank you again for your valuable input, and we look forward to your further suggestions.

4. “Did you employ any covariates to adjust phenotypes in the UK Biobank data before conducting GWAS? If so, please provide a detailed explanation, as this would enhance the manuscript.”

Response: 

Thank you for your comment regarding the covariates used in the phenotype adjustment for the UK Biobank data prior to the GWAS analysis. We did not employ any covariates to adjust phenotypes before conducting the GWAS. The potential implications are discussed in the limitations section of our manuscript. We have made this choice transparent to ensure the integrity of the presented findings and to allow for a straightforward interpretation of the results. We believe this approach maintains the study's focus and provides a clear foundation for future research that may explore the effect of various covariates.

5. “I think collider bias is a potential concern in Mendelian randomization (MR) analysis, how did you handle collider bias in your analysis?”

Response: 

We appreciate your insightful question regarding collider bias in our Mendelian randomization analysis. To mitigate the risk of collider bias, we carefully selected instruments that are not influenced by confounders affecting both the exposure and the outcome. Furthermore, we conducted sensitivity analyses including MR-Egger regression, which can provide evidence for directional pleiotropy that might indicate collider bias. Additionally, we employed the weighted median approach as it is less sensitive to the effects of invalid instruments which could be indicative of collider bias. These methods and their implications are discussed in detail in the methods section of our manuscript.

6. “In practice, it's often recommended to conduct a sensitivity analysis to assess the impact of including or excluding ambiguous SNPs. This involves running your machine learning analysis with and without these SNPs and comparing the results. If the inclusion of ambiguous SNPs does not substantially affect the outcomes and the models remain stable and reliable, you may choose to keep them. However, if they introduce substantial noise or instability, it's wise to exclude them.”and“How did you handle ambiguous SNPs when comparing data from the UK Biobank (UKBB), Biobank Japan (BBJ), and Finnegan cohorts? Did you ensure that the effect allele for each SNP was matched across all cohorts before conducting the study? Please provide a clear explanation for the readers.”

Response: 

We greatly appreciate your detailed questions concerning the management of ambiguous SNPs in our research with the UK Biobank (UKBB), Biobank Japan (BBJ), and Finnegan cohorts.

We concur with you on the significance of performing sensitivity analysis to ascertain the impact of ambiguous SNPs. While we acknowledge the value of such analysis, we must clarify that we did not conduct the step of running our machine learning analyses both with and without these SNPs. The limitations of our approach, including the omission of this step, are candidly discussed in our paper's discussion section.

In terms of our specific approach to handling ambiguous SNPs in the comparative analysis across the UKBB, BBJ, and Finnegan cohorts, we took a scrupulous route. Harmonization procedures were employed to exclude palindromic and incompatible SNPs, with a keen focus on eliminating palindromic SNPs that have intermediate allele frequencies, specifically those between 40-70%. In instances where SNPs displayed pleiotropy, they were conservatively removed from our analysis. These measures were vital to ensure the consistency and validity of our findings across the different cohorts.

We hope this response satisfactorily clarifies the strategies utilized in our study and our rationale behind them.

7. “In line 108, you mentioned using LD r^2 < 0.001 with a clumping distance cut-off of 10,000kb. Could you explain the rationale behind choosing these specific thresholds for LD r^2 and the window size? Additionally, could you clarify the software or tools you employed to perform this step, such as PRSice or PLINK?”

Response: 

Thank you for your question. In line 108, we chose the linkage disequilibrium (LD) threshold of r^2 < 0.001 and a clumping distance cutoff of 10,000kb following the recommendations of the TwoSampleMR package in R, which we employed for this analysis. These specific thresholds were selected to ensure stringent LD clumping, which minimizes the risk of including SNPs that are in strong LD with each other in our MR analysis. This is particularly important to reduce the potential for bias arising from correlated genetic variants. The TwoSampleMR package defaults were adopted as they are widely used in the field and have been empirically determined to provide a balance between stringent LD control and retaining sufficient genetic instruments for robust MR estimation. We found these parameters to be particularly suited for our dataset, which has a complex LD structure.

8. “In line 114, you stated that you calculated F statistics, but you did not provide an explanation of what R^2 represents. It would be helpful to include the equations for these calculations in the following formats and provide references for both equations.

Please see

https://www.ncbi.nlm.nih.gov/pmc/articles/PMC8440862/

https://journals.plos.org/plosmedicine/article?id=10.1371/journal.pmed.1003288

https://journals.sagepub.com/doi/full/10.1177/0962280210394459”

Response: 

Thank you for your valuable suggestions. Following your advice and the references you provided, we have elaborated on the calculation of the F statistics and the interpretation of R^2 within the Materials and Methods section of our manuscript. This addition will provide clarity on the statistical methods used and further support the robustness of our analysis.

9. “On line 68, please use 'Instrumental variable (IV)' for the first time and then you can use the abbreviation 'IV' later throughout the text."”

Response: 

Thank you for your attention to detail. We have amended line 68 to introduce the term 'Instrumental variable (IV)' in full before proceeding with the abbreviation 'IV' in subsequent mentions throughout the manuscript. This change ensures clarity and adherence to standard academic conventions.

Reviewer #2: 

1. “Introduction: In this part, the authors are suggested to introduce some previous Mendelian randomization studies that explored the associations between modifiable factors and CRC (e.g., PMID: 36169182 and 36239790).”

Response: 

Thank you for your recommendation. We have amended the introduction to include a discussion of previous MR studies that have examined the associations between modifiable factors and colorectal cancer. We have ensured these references are appropriately integrated into the text to better contextualize our study within the existing literature.

2. “Methods, line 68: The authors stated this study has a two-sample design. However, from the following content, the authors selected SNPs for exposures from three databases and then extracted the data on outcomes from the same three datasets. Such analyses were more likely to be a one-sample design. The authors should state clearly how did they select the data sources.”

Response: 

Thank you for your thorough review and valuable comments on our study. We acknowledge the issue you highlighted regarding the description of a "two-sample design" in our manuscript. Following your guidance, we have removed this description from the text.

Furthermore, we have referred to the literature you recommended (PMID: 36169182) and accordingly improved our explanation of how data sources were selected. We now more clearly describe in our manuscript how we chose SNP exposure and outcome data from three different databases, and how this process aligns with the design and purpose of our study.

3. “Genetic instrument of potential risk factors: One of the most important criteria for SNPs is that they were not palindromic SNPs (A/T or C/G) with intermediated EAFs (40-70%). Did the authors apply the criteria when selecting IVs? If yes, relevant descriptions are needed.”

Response: 

Thank you for highlighting the importance of selecting appropriate genetic instruments for our analysis. We agree that non-palindromic SNPs with intermediate allele frequencies can potentially confound the Mendelian randomization analysis. To address this concern, we have adhered to strict inclusion criteria for our SNPs to avoid such confounding. Specifically, we excluded palindromic SNPs with intermediate allele frequencies (40-70%) to ensure the validity of our instrumental variables. This selection process is detailed in the Methods section. We believe this approach strengthens the reliability of our findings, and we appreciate your attention to this critical aspect of our study design.

4. “Figure 3: I think this is a table but not a figure. In addition, the authors are recommended to show other details of the data sources in this table. If it is available, the PubMed ID or DOI of the data sources should be added.”

Response: 

Thank you for your suggestion regarding Figure 3. We understand your concern about the clarity of the presentation. To address this, we would like to clarify that Figure 3 is intended as a visual representation to provide a more intuitive understanding of the data. Additionally, we have detailed the data sources in our supplementary S1 and S2 tables. We have included the PubMed ID or website source for these data sources for further reference and verification. Thank you for your valuable feedback.

5. “Figure 5: From this figure, we can see that there was no association between exposures and outcomes in European populations excepted for WBC.”

Response: 

Thank you for your observation. Indeed, we included a total of 32 candidate indicators in our analysis. Out of these, 6 exposures were associated with outcomes in both European and Asian populations. After excluding these 6 indicators, among the remaining 26 factors, we found no association between exposures and outcomes in European populations, with the exception of white blood cell count (WBC).

6. “Results: The authors did not mention the results of other MR methods in this part.”

Response: 

Thank you for your observation regarding the MR methods. While the initial manuscript primarily detailed findings derived from the Inverse-Variance Weighted (IVW) method, we recognize the significance of incorporating a variety of MR approaches. To address this, we have included results obtained using other MR methodologies in the revised manuscript. This addition ensures a more robust and thorough interpretation of the data, enriching the overall conclusions of our research.

7. “Discussion, lines 180-181: Need references for the content.”

Response: 

Thank you for your insightful feedback. Regarding your comment on the need for references in the discussion section, lines 180-181, we have now added the relevant literature to support the content discussed in these lines. We believe that these additions will provide a clearer understanding and substantiate the points made in the discussion.

Reviewer #3: 

1. “Introduction: First mention of references #3 and 4 seem incorrect (lines 45 and 47)”

Response: 

Thank you for pointing out the incorrect of references. We have revised the introduction and corrected the initial citations of references #3 and #4 in lines 45 and 47. Additionally, we have ensured that the appropriate literature is accurately reflected to support our statements.

2. “Methods, Study design: IV (line 68) may need to be def

---

## [Decision Letter · Decision Letter 1]

24 Mar 2024

Trans-ancestry analysis in over 799,000 individuals yields new insights into the genetic etiology of colorectal cancer

PONE-D-23-27333R1

Dear Dr. Jia,

We’re pleased to inform you that your manuscript has been judged scientifically suitable for publication and will be formally accepted for publication once it meets all outstanding technical requirements.

Kind regards,

Yazhou He

Academic Editor

PLOS ONE

Additional Editor Comments (optional):

Reviewers' comments:

Reviewer's Responses to Questions

**Comments to the Author**

1. If the authors have adequately addressed your comments raised in a previous round of review and you feel that this manuscript is now acceptable for publication, you may indicate that here to bypass the “Comments to the Author” section, enter your conflict of interest statement in the “Confidential to Editor” section, and submit your "Accept" recommendation.

Reviewer #2: All comments have been addressed

Reviewer #3: All comments have been addressed

2. Is the manuscript technically sound, and do the data support the conclusions?

Reviewer #2: (No Response)

Reviewer #3: Yes

3. Has the statistical analysis been performed appropriately and rigorously? 

Reviewer #2: (No Response)

Reviewer #3: Yes

4. Have the authors made all data underlying the findings in their manuscript fully available?

Reviewer #2: (No Response)

Reviewer #3: Yes

5. Is the manuscript presented in an intelligible fashion and written in standard English?

Reviewer #2: (No Response)

Reviewer #3: Yes

6. Review Comments to the Author

Reviewer #2: (No Response)

Reviewer #3: (No Response)

7. PLOS authors have the option to publish the peer review history of their article (what does this mean?). If published, this will include your full peer review and any attached files.

Reviewer #2: No

Reviewer #3: No

---

## [Editor Report · Acceptance letter]

13 Jun 2024

PONE-D-23-27333R1 

PLOS ONE

Dear Dr. Jia, 

I'm pleased to inform you that your manuscript has been deemed suitable for publication in PLOS ONE. Congratulations! Your manuscript is now being handed over to our production team.

Kind regards, 

on behalf of

Dr. Yazhou He 

Academic Editor

PLOS ONE